# Towards the Implementation Process of Accessible Virtual Campuses in Higher Education Institutions in Latin America

**Francisco Sánchez Vásquez [1],\*, Juan Carlos Pérez-Arriaga [1], Gerardo Contreras Vega [1], Sergio Luján-Mora [2] and Salvador Otón Tortosa [3]**

[1] Facultad de Estadística e Informática, Universidad Veracruzana, Xalapa 91020, Mexico; juaperez@uv.mx (J.C.P.-A.); gcontreras@uv.mx (G.C.V.)

[2] Department of Software and Computing Systems, University of Alicante, 03690 Alicante, Spain; sergio.lujan@ua.es

[3] Department of Computer Science, University of Alcalá, 28805 Alcalá de Henares, Spain; salvador.oton@uah.es

\* Correspondence: fransanchez@uv.mx

**Abstract:** Ensuring equitable and inclusive access to educational services in Higher Education Institutions (HEIs) requires the development of strategies that consider the diversity of their academic members, administrative staff, and students, as well as the use of information and communication technologies. The identification of requirements for technological accessibility in HEIs allows for the establishment of actions aimed at considering accessibility aspects in the processes of admission, permanence, and graduation, in order to support students with disabilities in their transit through these institutions. Having a systematic approach to guide the design of educational strategies in HEIs contributes to the identification of areas for improvement for the benefit of educational quality and community members. This article describes the proposal of a process based on the Plan-Do-Check-Act (PDCA) cycle, and a methodology for the implementation of accessible learning environments oriented to the implementation of an accessible virtual campus based on the establishment of five defined phases: diagnosis, planning, implementation, control, and tracing. This proposal is aimed at supporting Latin American HEIs in the integration of technological accessibility requirements from a systematic and continuous improvement approach.

**Keywords:** accessibility; higher education; PDCA cycle; accessible virtual campus

## 1. Introduction

The increase in the use of Information and Communication Technology (ICT) contributes to the creation of changes that allow the digitalization of processes within organizations. Higher Education Institutions (HEIs) are no exception to this. Therefore, for this digital transformation to occur in universities, we must consider structural changes to the entire institution. The restructuring includes how the institution manages academic and administrative personnel and students [1].

The sustainable development goals of the United Nations, in particular goal 4, which refers to ensuring inclusive and equitable quality education, emphasizes the need for carrying out actions to ensure equal access for men and women at different levels of education, including undergraduate studies, by 2030. Additionally, it seeks to eliminate gender inequality and ensure equal access to education for all vulnerable groups (people with disabilities, indigenous groups, and children in vulnerable conditions) [2].

Actions at different levels of an HEI are necessary to create academic processes that fulfill the requirements of people with disabilities [3]. These actions should involve the processes of admission, permanence, and graduation, so that students, especially those with disabilities, have the necessary support during college.

Molina-Perez, J.–Pulido-Jimenez, C. [4] mentions the existence of limitations that negatively impact the implementation of ICT on pedagogical environments and can represent an obstacle in the implementation of accessible campuses. These limitations include the following: the lack of appropriate content, lack of support or training for staff, and the lack of digital skills on the part of students and teachers. Due to this, having a mechanism to determine the technological accessibility requirements in HEIs ensures positive results to achieve the implementation of accessible virtual campuses by enabling better decision making.

The International Organization for Standardization (ISO) defines accessibility as the extent to which a system, product, service, environment, or facility can be used by the highest number of users possible, regardless of their characteristics, abilities, and requirements, while still allowing it to meet its intended goals for which it was designed [5,6]. Thus, having strategies supported by accessible technology helps to reduce existing digital barriers related to access of academic services for people with disabilities [7].

The concept of technological accessibility refers to technology designed to be adapted to the user's requirements, and to make itself accessible without requiring external assistive technology. Furthermore, a technology is accessible in the case that it is compatible with assistive technologies [6]. Accessible technology, also called adaptive technology, is designed to cover most user abilities, thus it is sometimes referred to as customizable technology [5]. Accessible technology encompasses the creation of websites, documents, digital resources, and other features that conform to the accessibility requirements that allow for availability to a wide variety of users [8].

With regards to education, United Nations Educational, Scientific and Cultural Organization (UNESCO) [9] establishes that it is the responsibility of the state to promote inclusive and equal education at all levels, including higher education.

In certain countries, strategies have been developed in order to promote technological accessibility, mainly in the development and/or adaptation of learning management systems (LMS). However, it should be noted that there is a clear need for the generation of accessible educational material that allows students with disabilities to have the necessary elements for their professional development [10].

Since 2000, in countries like Mexico, the educational requirements of people with disabilities have been identified due to the creation of legal frameworks focused on safeguarding the rights of these people, leading to an increase in enrollment at different educational levels [11].

For Behm [12], a university campus is linked to the concept of spatial analysis. In this sense, the university campus does not focus on a concept of buildings, but represents the administrative, academic, and physical processes that involve the student community within HEIs. Implementing a virtual campus in an HEI requires a multidimensional analysis that begins from an organizational perspective, and considers faculty and staff training processes, curricular restructuring, student support, content generation, and other aspects. On the other hand, it should be noted that the use of accessible platforms for learning management is a priority when it comes to monitoring student performance in an educational program.

The lack of attention to the educational requirements of people with disabilities by HEIs sometimes leads to legal problems in the institution. In addition, it represents an obstacle for the academic development of students, affecting their professional careers [13–16]. Additionally, events such as the COVID-19 pandemic evidence the lack of strategies aimed at promoting the accessibility of academic processes in HEIs.

It is recommendable that virtual inclusion strategies consider the combination of aspects such as pedagogical, technological, psychological, among others. The results described by Reyes, J. et al. [17] evidence that considering accessibility requirements is an important factor to promote the inclusion of students with disabilities. Online learning strategies that consider accessibility facilitate the active participation of students in learning processes.

The work of Reyes, J. et al. [17] concludes that it is necessary to evaluate the effectiveness of inclusive practices that combine virtual education. On the other hand, by adopting systematic approaches, it is possible to define quality improvement programs that guide the implementation and evaluation of accessible learning strategies. Asif and Raouf [18] emphasize that the lack of systematic implementation could contribute to the existence of isolated practices from institutional or organizational processes.

Therefore, having a continuous improvement process that allows HEIs to detect areas of opportunity in terms of accessibility, promotes the increase in educational quality and actions that are in accordance with the educational requirements of students with disabilities [19].

There are efforts focused on the integration of accessibility in educational processes in HEIs. Caforio [20] proposes the use of a framework that contributes to the accessibility of the learning offer in European universities; however, this framework focuses on institutional policy indicators, course design, as well as the publication and evaluation of these.

In Lowenthal's work [21], frameworks for quality assurance from an accessibility perspective are analyzed, in which the Open SUNY Course Quality Review Scorecard (OSCQR) highlights the use of guidelines for text formatting, color contrast, captioning, among other elements. The study mentions that although accessibility is a topic of interest in HEIs, the actions to increase accessibility focus on physical barriers, on the design of accessible content, and on the accessibility of technological platforms.

Working on accessibility strategies in HEIs from a systematic approach that considers institutional aspects such as strategic planning, stakeholders' engagement, human resources, as well as admission, permanence, and graduation processes, contributes to guaranteeing equitable access for all [22]. This document describes a proposal of a process based on the Plan-Do-Check-Act (PDCA) cycle [23], and a methodology for the implementation of accessible learning environments oriented to the implementation of an accessible virtual campus, based on the establishment of five defined phases: diagnosis, planning, implementation, control, and tracing. This proposal is aimed at supporting HEIs in their integration of technological accessibility requirements based on a systematic and continuous improvement approach.

This paper is divided into the following sections: Section 2 describes both objectives and the research method; Section 3 presents the results of a multivocal review of the literature on technological accessibility requirements in HEIs, as well as the regulations or policies associated with the generation of strategies in order to respond to the requirements detected; Section 4 shows the results of a multivocal review on accessible virtual campuses and their status in HEIs in order to identify practices that support the implementation of virtual campuses; Section 5 describes a proposal of a process to guide the implementation of an accessible virtual campus. The process is based on the identification of accessibility requirements and the adaptation of methodologies for the implementation of accessible learning environments in accordance with the stages of the PDCA cycle. Section 6 presents the conceptual validation of the proposal by experts. Finally, Section 7 presents the summary of this study, and explores opportunities to consider for future work.

## 2. Research Methods

The research is oriented to the definition of a process that guides the systematic implementation of accessible virtual campuses in HEIs, based on the identification of technological accessibility requirements and practices documented in the literature that relate to the implementation of accessible virtual campuses. Multivocal literature reviews were conducted in order to identify technological accessibility requirements, as well as the practices implemented by HEIs. These reviews include both white literature as peer-reviewed papers and grey literature (Appendix A, Appendix B, Appendix C, Appendix D), which uses whitepapers, web pages, and technical reports, among other sources of information. The use of multivocal reviews allows the researchers to include within their search results

the body of knowledge reported by professionals, practitioners, and educational institutions, among other authors in non-formal sources [24]. The conducted reviews include non-Latin American countries such as Spain, as well as the United States of America, as a result of their work on policies and laws related to web accessibility.

The implementation of improvement processes in HEIs requires a systematic approach that includes human resources, information resources, procedures, regulations, and infrastructure [25]. The PDCA model provides a framework for quality assurance of academic processes in HEIs. In the PDCA model, the definition of strategic plans is established as a starting point (Plan) to subsequently execute the activities identified in the planning phase (Do); make an analysis of the current situation of the results that are being obtained and detect elements that allow continuous improvement (Check); finally, through the evaluation strategies of the strategic plans, the areas and activities of improvement (Act) are defined. Figure 1 describes the research method.

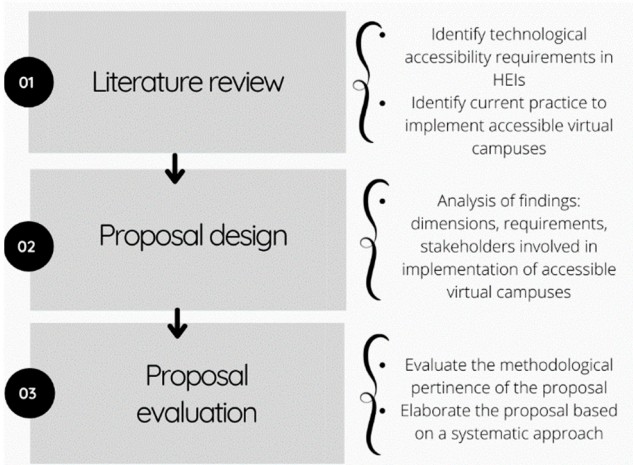

**Figure 1.** Research methods.

### 3. Technological Accessibility Requirements in Higher Education Institutions

Due to technological growth, virtual platforms have been used as self-sufficient teaching media to provide services for distance learners which they can interact with. Although there are technological platforms that provide distance educational spaces, it is necessary that the requirements of students with disabilities are considered in order to create accessible virtual environments [26,27].

Some countries in Ibero-America have focused on enacting standards, public policies, or internal regulations [4,10] that provide legal elements to support the rights and conditions of people with disabilities, so that people do not feel excluded in their environment and can develop their academic activities on an equal footing [28,29]. Identifying the requirements for technological accessibility in HEIs contributes to the creation of flexible educational environments that are adapted to the conditions required for students to access the services necessary for their academic training.

### 3.1. Identification of Technological Accessibility Requirements in Environments Related to HEIs

The multivocal review [30] describes the technological accessibility requirements in HEIs based on the identification of institutional policies, application of strategies, regulations, and guidelines that support the actions carried out. In addition, the results obtained by HEIs in meeting requirements for technological accessibility are documented.

The findings described in this section are an excerpt of the review conducted by Vazquez et al. [30], due to its relevance to the research. As an extension to this review, a thematic map is added to show the review results and their associations.

*3.2. Results Obtained*

Countries such as Puerto Rico, Ecuador, Colombia, Argentina, Spain, and others have developed strategies aimed at identifying requirements for technological accessibility.

Figure 2 shows the results of the studies found by country that respond to one or more of the research questions. The studies considered are presented in Appendices A and B.

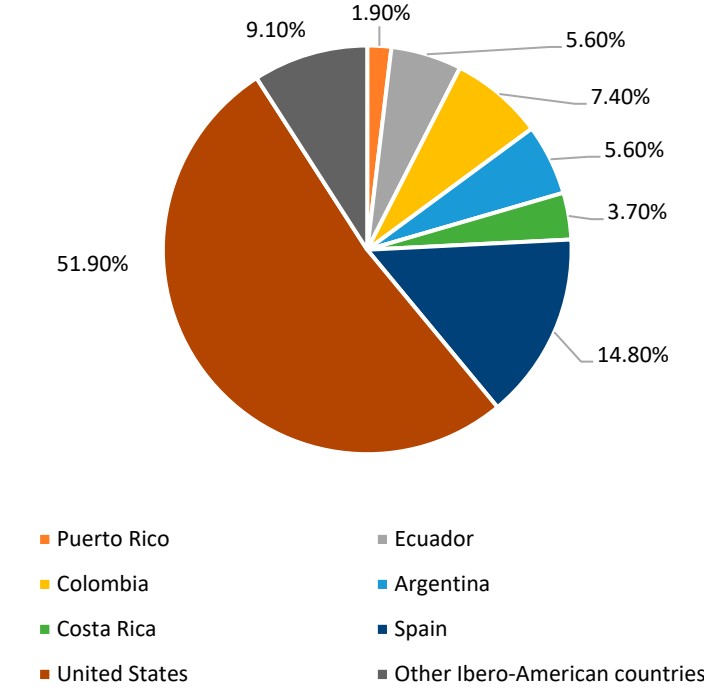

**Figure 2.** Studies found by country related to accessibility requirements and technological strategies.

*3.3. Answers to Research Questions Related to Technological Accessibility Requirements in HEIs*

3.3.1. NRQ1 What Are the Actions, Plans, or Policies Related to Technological Accessibility That Are Being Implemented in HEIs Reported in Literature?

There are different elements that distinguish disability and adaptations that are required to support students with disabilities in their academic efforts. For instance, there are requirements that imply the design of mechanisms aided by technology to overcome educational barriers. On the other hand, the application of existing norms with accessibility in mind, can help promote the matter [31].

The actions identified focus on the adaptation of physical spaces and the implementation of accessible classrooms. Furthermore, internal information technology (IT) teams or departments within HEIs with specialized personnel to train and guide users in the institution towards the development of accessible content have been identified. Laws and norms have been established in order to promote the access to education. As a result, many HEIs have issued institutional policies that commit support for people with disabilities, personnel training, and for research into accessibility issues. Implementing institutional policies that include accessibility aspects contributes to the development of accessible technological platforms as a response to the requirements of students with disabilities.

### 3.3.2. NRQ2: What Are the Documented Norms in Literature That Regulate Aspects of Technological Accessibility in HEIs?

Currently, international, national, or autonomous bodies exist in each country that are in charge of promoting the equality and integration of individuals with disabilities, while safeguarding fundamental human rights of all people. Bodies like UNESCO, Convention for Human Rights, and the Organization for Economic Co-operation and Development (OECD), are international entities that promote and are in charge of ensuring compliance with accessibility normativity within member nations [9,32–34].

In Mexico, some of the organizations focused on promoting improvements in the field of education, research, services, transportation, and development for people with disabilities include the National Association of Universities and Higher Education Institutions (ANUIES), and the Department of Public Education (SEP) [35,36]. In addition, legislations and standards exist that guarantee compliance with accessibility requirements in the field of IT, including products, services, hardware, software, and web content under the country's jurisdiction.

In the United States of America, laws and regulations have been passed that prohibit discrimination against people with disabilities, mandate web content be accessible by following standards in combination with undergoing compliance inspections, and increase accessibility in academic spaces. Similarly, countries such as Ecuador, Chile, Argentina, and Spain contain legislation that promotes and guarantees accessibility. Colombia, Puerto Rico, and Peru have each made strides to guarantee education to people with disabilities [33,37–39].

### 3.3.3. NRQ3: What Are the Requirements Addressed in Technological Accessibility Strategies Implemented by HEIs?

Among the requirements identified in HEIs that relate to access barriers to adequate educational content and transportation for students with disabilities are the implementation of policies, norms, and syllabi about accessibility topics; the elaboration of strategies for inclusive environments; the training of personnel in the areas of accessibility; and the implementation of classrooms and physical spaces with adaptability in mind [1,27,40–43].

Student desertion for those with disabilities enrolled in HEIs is a consequence of lack of adequate spaces, accessible technology, materials, and technological resources. These indicators form part of the detected strategies that seek to bring the required support during college.

Actions that support the technological development of accessible virtual platforms in HEIs include the founding of IT departments for the purpose of developing of virtual platforms, and the digitalization of educational material while complying with accessibility standards [44]. For instance, ISO/IEC 40500 and the Web Content Accessibility Guidelines (WCAG) 2.0 [45,46] by the World Wide Web Consortium (W3C) are a set of popular guidelines for the creation of accessible websites that focus on the development of systems designed to include people with disabilities as users.

There are strategies oriented towards attending to the lack of sensitized personnel, pedagogical barriers, and attitudes that limit the integration of students with disabilities to a college environment. Identifying requirements is not enough to foster accessible learning environments. Nonetheless, this can be helpful for detecting points for evaluation to determine accessibility levels within an educational institution, thereby promoting improvements that tear down existing barriers and facilitate the integration of students with disabilities into the HEI [43].

### 3.3.4. NRQ4: What Are the Results Documented in the Literature That Support the Functionality of Technological Accessibility Strategies Implemented by HEIs?

By implementing technological accessibility strategies in HEIs, positive results are observed, such as a tendency to enhance the integration of students with disabilities and the quality of services offered to them. One of the benefits identified in literature includes the use of accessibility techniques for online courses by adding subtitles to audio elements and alternate texts to visual elements for the understanding of those with sight or hearing disabilities.

Involving students with disabilities in the processes that compose an educational institution tends to increase adaptability and accessibility in physical spaces as well as learning material; as a result, this fosters inclusive environments within classrooms. Commissions and groups are created that contribute to the strengthening of strategies, sensitivity training, employee training, and equality. Other positive impacts identified are a decrease in discrimination, alongside an increase in inclusive counseling and physical spaces for those with disabilities in the student community. These resulting benefits serve to better integrate this vulnerable minority into academic activities.

### 3.4. Discussion Results

In order to elaborate a thematic synthesis, the Cruzes et al. [47] method was followed. This consists of defining relevant terms based on the main topic of the research, and finding their connections with other extracted data that represent a relevant topic in the systematic review of the literature.

In Figure 3, a thematic map can be seen in which concepts associated with technological accessibility have been identified. Each concept contains different elements derived from the studies obtained from the investigation.

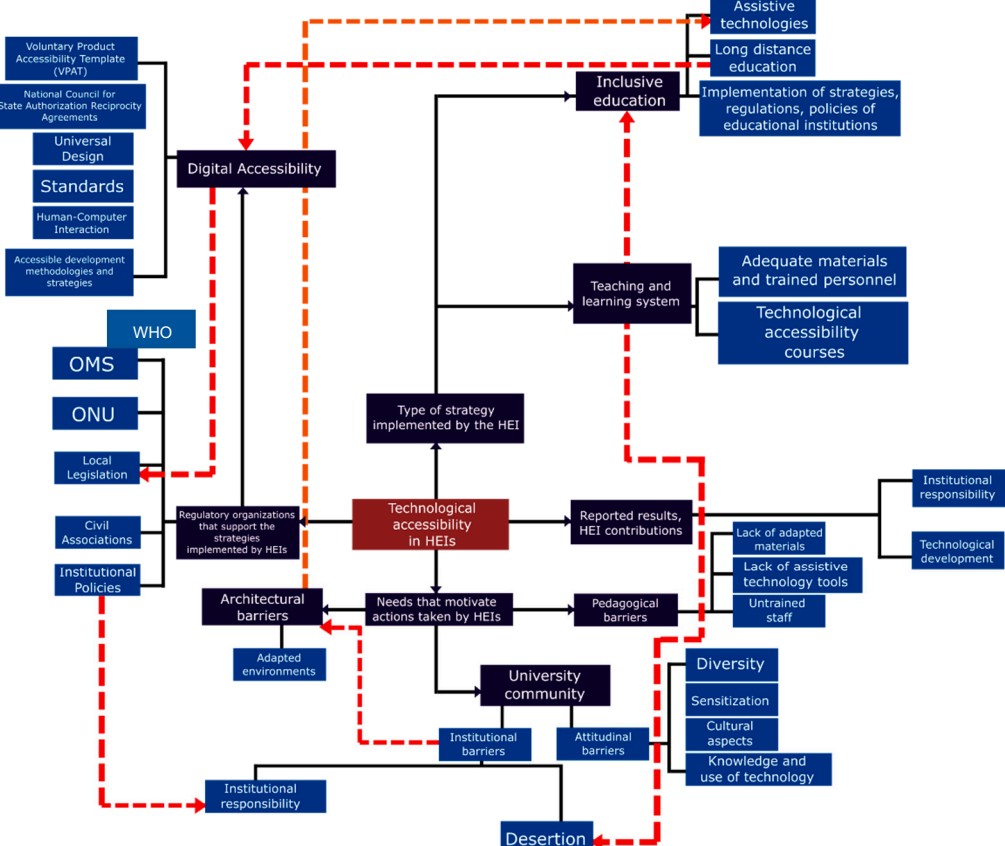

**Figure 3.** Thematic map of association of terms in response to the research questions [47].

Based on white literature (Appendix A) and grey literature (Appendix B) studies, specific concepts were identified as barriers that represent obstacles to the implementation of accessible virtual campuses. Some of these barriers include the following:

- **Architectural barriers**. Manifested by the absence of accessible classrooms, accessibility signage and access ramps, among others.
- **Pedagogical and human resources barriers**. Defined by the lack of sensitivity training, employee training, and for educators in accessibility themes.
- **Attitude barriers**. Encompassed by discrimination towards people with disabilities on the part of the campus community, disregard for norms and regulations, as well as cultural aspects or beliefs.

Additionally, strategies focused on attending to the accessibility requirements identified include policy making; implementation of technological accessibility courses; personnel training; specialized learning materials; and educational resources made available by ICT.

As far as laws and regulations that support the creation of accessible technology strategies are concerned, there exist organizations with objectives to enforce norms, such as governmental bodies and legislation passed within each country. Among the countries identified that have passed disability laws and acts, the United States of America leads as the primary promoter in the application of accessibility norms within its institutions, followed by nations such as Spain, Mexico, Colombia, Chile, Argentina, Ecuador, Brazil, and the U.S territory of Puerto Rico.

With reference to web accessibility guidelines, adherence to the standards by the W3C such as WCAG and WAI-ARIA stand out, as they contain principles and recommendations for digital content creation in websites, interface design, and the application of methodologies for the development of accessible software.

## 4. Accessible Virtual Campuses

Based on a multivocal review of the literature, we sought to identify the current state of art of accessible virtual campuses in HEIs, as well as their characteristics. In the review, the defined objectives of the search are based on locating existing initiatives to create virtual campuses and associated methodologies to implement them. These research objectives are as follows:

- Identify the actual status in accessible virtual campuses or similar environments for e-learning in an HEI.
- Document the findings related to characteristics on accessible virtual campuses.
- Provide a global vision of contemporary initiatives for the implementation of virtual campuses and e-learning in HEIs.

### 4.1. Accessible Virtual Campuses

In keeping with these research objectives, the following research questions (RQ) have been defined:

**RQ1**: Which HEIs have accessible virtual campuses?

**RQ2**: What aspects of accessibility are considered in the design of a virtual campus in an HEI?

**RQ3:** What are the requirements addressed in technological accessibility strategies implemented by HEIs?

**RQ4**: What are the results documented in literature that support the effectiveness of technological accessibility strategies implemented by HEIs?

*4.2. Search Process*

The search string used in the multivocal literature review was:

*("Virtual Campus"|E-campus|"Virtual Education"|"Virtual University") AND ("Accessibility Guidelines"|"Accessibility Standards") AND ("Higher Education"|Universities|University|"Higher Education Institution"|HEI) AND (Disability|Disabilities|Impairment) AND (Accessibility|Accessible) AND (Adaptability|"Adaptive System"|Adaptable) AND (Implementation Evaluation|Proposal Measurement)*

In order to carry out the search, it was determined to use four search engines: Google Scholar, ACM Digital Library, IEEE Xplore, and ERIC. Moreover, grey literature available via Google searches was included as a source of information.

The literature was filtered through the application of the following inclusion criteria (IC):

- IC1: Articles published between 2015 and 2021.
- IC2: Articles written in either English or Spanish.
- IC3: Articles about technological accessibility strategies.

The exclusion criteria (EC) contemplated were the following:

- EC1: The paper is not related to standards, norms, plans, or actions, even if the paper alludes to technological accessibility.
- EC2: Access to the full paper was not possible.

*4.3. Results Obtained*

At this stage, investigation followed the search string. The results found are shown in Figure 4.

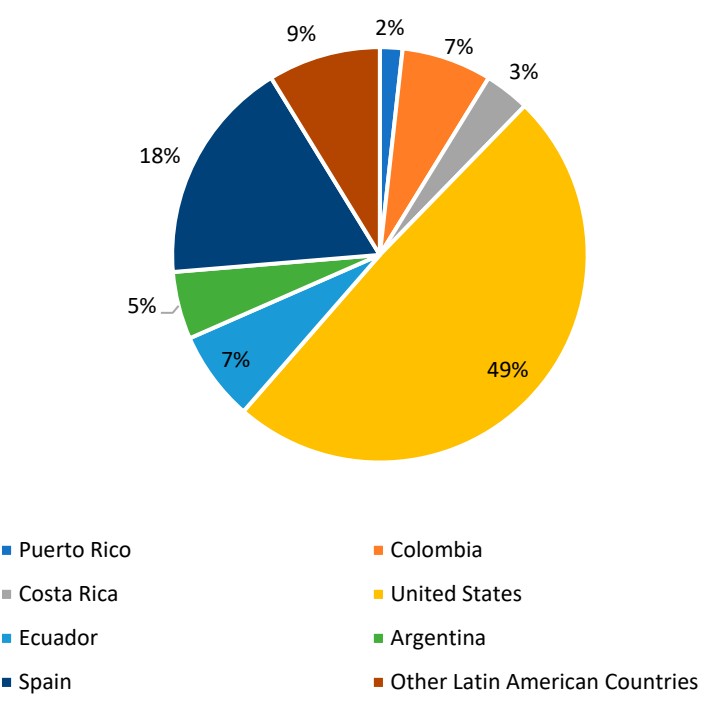

**Figure 4.** Studies found by country related to accessible virtual campuses.

After applying the inclusion and exclusion criteria, the results yielded 62 documents relevant to the investigation.

*4.4. Answers to Research Questions Related to Accessible Virtual Campuses*

4.4.1. RQ1: Which HEIs Have Accessible Virtual Campuses?

Reports mention that institutions have employed strategies in order to increase the quality of education or to detect the requirements of their students. As a pioneer from Iberia, the University of Lisbon, reports improvements to their application processes by the introduction of an admission process, offering students help through tutoring and trained personnel until graduation by enforcing policies in line with international standards. Nonetheless, the university delivers administrative processes that lack adaptability [48].

Other universities have focused on identifying the requirements of students and implementing accessibility elements in physical spaces. Such is the case for Open University UK which determined that the early detection of requirements positively influences the creation of digital resources adapted for multiple disabilities [49]. These digital resources can then be utilized to create a virtual platform adequate for all.

With regards to accessibility standards and guidelines for websites, the University of La Rioja and the University of Central Venezuela have begun to develop their own virtual platforms through adherence to WCAG [50,51]. For the construction of virtual platforms with accessible interfaces, use of a universal design has been proposed. This universal design would be applied to e-learning concepts and the elaboration of learning objects adapted for content for people with disabilities [52].

4.4.2. RQ2: What Aspects of Accessibility Are Considered in the Design of a Virtual Campus in an HEI?

In order to achieve the implementation of a design appropriate to the multiple characteristics of students, there is a model that applies a questionnaire and provides indicators showing which deficiencies exist in the institutional processes and thus require adaptation [41]. Among the indicators that need to be considered for an adequate design of virtual campus, some are the following:

- Design tools to use information about the respondent and the context in order to aid comprehension.
- Develop automated approaches to identify relevant accessibility issues.
- Evaluate the accessibility and representation of feedback tools through analysis.
- Conceptualize development and user changes over time as part of accessibility processes.
- Develop a focus on socially accessible designs for interaction.
- Design mechanisms so that feedback is relevant, receptive, and reflective.
- Consider the relationship between users and the organization.

These indicators seek to promote inclusion in systems and services that improve virtual platforms [49]. Another element to consider when creating accessible conditions in virtual platforms is to present alternative texts and subtitles to the visual elements that the websites present. Some reports state that this consideration has facilitated comprehension by the general student population, including people with disabilities [52,53].

In order to help students with visual and hearing disabilities, e-learning platforms include adapted video resources, audio descriptions, captioning, sign language, and long descriptions of images which are reproduced as audio speech.

In order to guarantee accessibility, it is common to consider the WCAG 2.2 guidelines in order to develop models that contemplate different disability categories such as visual, auditory, cognitive, motor, elderly, and linguistic.

In the development of MOOC design, there are resources such as accessible PDF text, audios and videos with subtitles, videos with sign language interpretation, alternative text in images, graphics and labels on resources, and descriptions of icons and symbols used in the course.

### 4.4.3. RQ3: What Are the Requirements Addressed in Technological Accessibility Strategies Implemented by HEIs?

A part of the requirements addressed in HEIs is satisfied through standards that consider accessibility elements for the development of educational products, such as learning objects and those established by the WCAG [43].

As part of the strategies implemented to attend to accessibility requirements, models and methodologies have been created in order to analyze, detect, implement, adapt, and evaluate the resources at hand, particularly available learning objects. These strategies focus on applying improvements in order to properly adapt to the various characteristics of users, including disabilities, and the use of virtual platform interfaces by this vulnerable group [34].

An important point for the design of accessible conditions within virtual platforms is the use of a LMS that features additions that can enhance interface interactions or the presentation of information on screen. Information presentation can be adjusted by changing font sizes, increasing color contrasts, adding subtitles, and adding speech synthesis, among other elements that aid students (especially those with disabilities) [34,54]. In order to include such features, models have been proposed that make use of an LMS or Massive Online Open Courses (MOOC) system to develop an accessible virtual platform [55]; in addition, such standardized formats have been proposed in order to produce an accessible universal design, with the goal of achieving simple navigation and information comprehension on a website [56].

The Faculty Playbook developed by Keefe et al. (2020) describes a set of guidelines to be used for remote teaching and online learning. This work provides guidance for course design and delivery. Individuals and institutions can find best practices that consider individual needs in course design. Additionally, the work mentions that the effectiveness of practices is improved when actors, such as course designers, faculty, institutional centers, staff, and support offices, are involucrate with the design of learning strategies. Also, it evidences the necessity to establish a continuous improvement approach for both course level and instructional evaluation, in order to maintain best practices in online course design, teaching, and learning.

### 4.4.4. RQ4: What Are the Results Documented in Literature That Back the Effectiveness of Technological Accessibility Strategies Implemented by HEIs?

Strategies were identified that implement manual and automatic tests that involve users. These are focused on receiving sufficient feedback in order to evaluate what improvements need to be implemented, or if the accessibility elements in place meet the requirements of users [54]. Among the strategies applied that support the creation of accessible virtual campuses are standards such as WCAG 2.0, in order to develop accessible websites. Through the implementation and development of websites that apply robustness, perception, operation, and pedagogical elements generally considered in accessibility standards, they present improvements in the educational learning of students with and without disabilities [57].

In order to guarantee that the development complies with established criteria, there are educational quality evaluation metrics with accessibility parameters which allow an HEI to obtain a quantitative value to adapt improvements in areas that require them [58]. Other ways to obtain a result pertaining to accessibility is by applying metrics and questionnaires to students with disabilities. This has changed the perspective in a positive way for users who interact with technology, and allows the application of improvements to administrative, educational, or regulatory processes involved [50,59].

Finally, strategies that help improve staff awareness and attitudes involve training teaching staff in technological accessibility and disabilities, in order to adapt educational material and provide a pedagogical quality service; this breaks down educational barriers, and promotes participation by students with disabilities [24].

In order to report the quality of an accessibility strategy, Batanero-Ochaita (2020) made students answer a seven-item questionnaire that collected their opinions about the ease of use and usefulness of the adapted platform. The instruments used to make this were a five-point Likert-scale form to guarantee the results they used, and Cronbach's alpha reliability coefficient to assess the consistency of the scale used.

The use of automated accessibility tools and testing with users are methods to measure the quality of technological solutions. Through the organization of focus groups, the obtained users' feedback is used to improve the accessibility of the platforms.

Nieves et al. (2019) show the validation of MOOC courses by implementing usability tests with users with and without visual functional diversity, and validating the information with a questionnaire that is designed under the principle of universal design, accessibility, and usability. The questionnaire was validated with the Delphi method by experts.

### 4.5. Discussion of Results

There is an increase in the efforts made by HEIs, such as implementing strategies to consider regulations in order to generate accessible environments, and adapting physical spaces and curricula; however, at the Latin American level it is possible to see that universities are behind in the implementation of strategies to contribute to the creation of accessible virtual campuses. Some factors that influence the lack of sufficient conditions are the inadequate economic resources, in addition to a lack of trained personnel, which represents a problem in terms of disparity. In other more developed countries, such as the United States, implementing accessible virtual campuses is done by following accessibility standards, such as those proposed by the W3C.

Despite having standards that imply universal design focused on accessible elements, it generally requires time to achieve adaptations and reach the goal of covering the requirements that arise from students, or adapting characteristics that still require attention to achieve accessible physical spaces and virtual platforms that involve the processes of the institution. Unlike Latin America, in the European region there is a constant increase in implementing strategies that comply with regulations and technological aspects, which helps students function naturally in the physical spaces and digital platforms of HEIs.

By satisfying requirements with the development of accessible virtual platforms, teaching classrooms have been installed that attempt to break down technological and pedagogical barriers within the HEI; however, the development requirements need to be evaluated through metrics, interviews, evaluation instruments or questionnaires that gather information, in order to improve the interaction on the virtual platforms based on their experience and activities within the virtual campus.

The research findings focus on the technical evaluation of existing tools, while disregarding processes or methodologies for the development of an accessible virtual campus. As such, there is a need for a suitable process for the creation of a virtual campus. Such a process should have its steps rooted in accessibility principles, in order to achieve an accessible virtual campus of high quality.

## 5. Process for the Implementation of an Accessible Virtual Campus

The use of ICT in the academic context can mitigate barriers that limit the participation of students with disabilities in teaching–learning processes. The accessibility requirements detected, such as lack of training of teaching staff, lack of adapted materials, in addition to scarce consideration of administrative processes in the design of inclusion strategies through ICT, highlight the importance of establishing systematic processes in order to guide the proper implementation of such strategies.

The correct implementation of accessible virtual campuses must be guided by a process that guarantees the quality of the teaching–learning process, beginning with the analysis of the accessibility status of the institution, and subsequently allowing the appropriate design of strategies to be followed in such implementations. Additionally, it is

convenient to include administrative processes in the design of strategies, since it is through them that students have access to academic information, as well as to the services provided by the HEI.

HEIs face a challenge when attempting to effectively respond to changes in the environment to offer quality services. The establishment of processes in organizations requires periodic evaluation in order to identify areas for improvement. This process-based approach establishes a systematic means to identify and manage processes within an organization [60]. The quality of educational services should not be measured by the quality of the final product, but instead it should be measured by the execution of the entire process. Quality in education should consider aspects such as availability of teachers, infrastructure, curriculum, and equipment, as a defined teaching–learning process [61].

Jain [61] mentions that there are two aspects related to quality in education. The first one sees the quality of the educational system as an entire system that includes schools, teaching–learning environments, policies, and other elements. The second aspect of quality of the educational system is related to what the system offers to the members of its community (students and teachers) from the perspective of the teaching–learning process. In the context of HEIs, the implementation of process-based educational models has been documented, where the aim is for the processes to function properly. The process-based quality educational model includes management, teaching, and learning processes [25].

PDCA is a cyclical tool that can be used as an intermediary to organize processes and systems, according to ISO 9001:2015 [62]. Each stage is described as follows:

- **Plan**: The system objectives, processes and resources needed to deliver an outcome in accordance with the organizations requirements and policies to manage risks and opportunities should be established.
- **Do**: This stage focuses on implementing what was established in the Plan phase.
- **Check**: The processes and products resulting from the implementation are supervised (in the areas where is required), verifying that the objectives, requirements, and planned activities are being carried out according to the plan and reporting the results.
- **Act**: The focus on taking measures in order to improve current performance.

Asif and Raouf [18] propose a framework inspired by the PDCA process model focused on assessing quality in higher education. The proposal mentions that quality assurance includes the development of strategic plans (plan), the execution of these plans in the (do) stage, collecting feedback from stakeholders such as students, graduates, staff, among others (check), and consolidating the stakeholders' feedback into the HEI that support the continual improvement in the (act) stage.

### 5.1. Methodologies for the Implementation of an Accessible Virtual Campus

In the multivocal literature review conducted, there were methodologies identified that consider the different organizational levels in HEIs with respect to the implementation of virtual learning environments. Hernández-Otálora et al. [63] proposes the systematic creation of a virtual learning environment as a set of defined phases and dimensions that enable the implementation of virtual spaces for learning. Phipps and Kelly [54] employ a holistic approach to e-learning founded on usability, accessibility, local factors, infrastructure, and available learning results. Lastly, Salvatierra [34] proposes criteria that should be considered for the creation of accessible e-learning platforms. These following criteria are:

1. **Establish Consciousness**: learn the requirements of users, the uses of resources, and approaches towards inclusivity.
2. **Research**: identify usability best practices for accessible platforms and content.
3. **Comprehension**: evaluate the adaptation of current practices and their potential application as learning objects.

4.  **Implementation**: execute changes based on current and researched practices. Work on adjustments as they are needed, in addition to search for possible alternatives and enhancements.
5.  **Evaluation**: verify the quality as well as the effectiveness of learning objects used by students.

In the methodologies described, some common elements stand out, such as discerning the requirements via an analysis or diagnosis; pinpointing practices that increase usability and accessibility in learning environments; making necessary adjustments; and the evaluation of steps taken towards the objectives and requirements established.

When assessing the processes involved in HEIs, a wide array of factors and stakeholders must be considered in order to balance the multiple requirements identified and generate a high-quality virtual campus [64].

### 5.2. Proposal of a Method to Implement an Accessible Virtual Campus

The proposal for a model to implement accessible virtual campuses emerges from the combination of the PDCA model for the continuous improvement of processes and products, and some of the deliverable indicators proposed by the Hernández-Otálora et al. [63] methodology for the implementation of accessible learning environments. In Figure 5, the integration of these two methodologies can be seen alongside the considered phases.

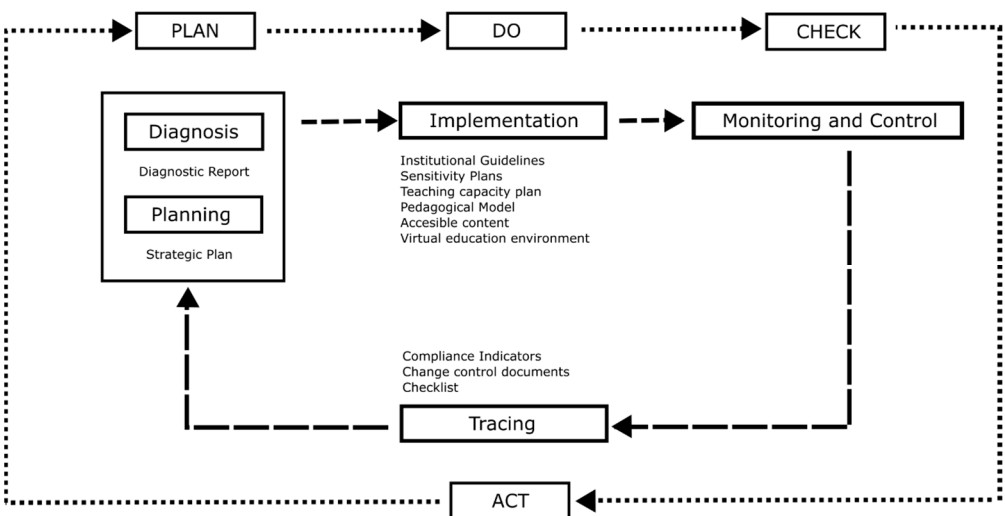

**Figure 5.** Integration of PDCA to the proposal based on the methodology of Hernández-Otálora et al. [63].

The following descriptions cover each phase of the proposal.

### 5.2.1. Phase 1—Diagnosis

In this diagnosis phase, the first step consists of defining the team that will oversee the analysis of initial accessibility conditions within the HEI. The objective of this phase is to establish a framework that meets the requirements of people with disabilities. Furthermore, in this phase the work to be done in each of the dimensions consisting of a virtual educational space is assigned; these are the organizational dimension, pedagogical dimension, academic dimension, and technological dimension (Figure 6).

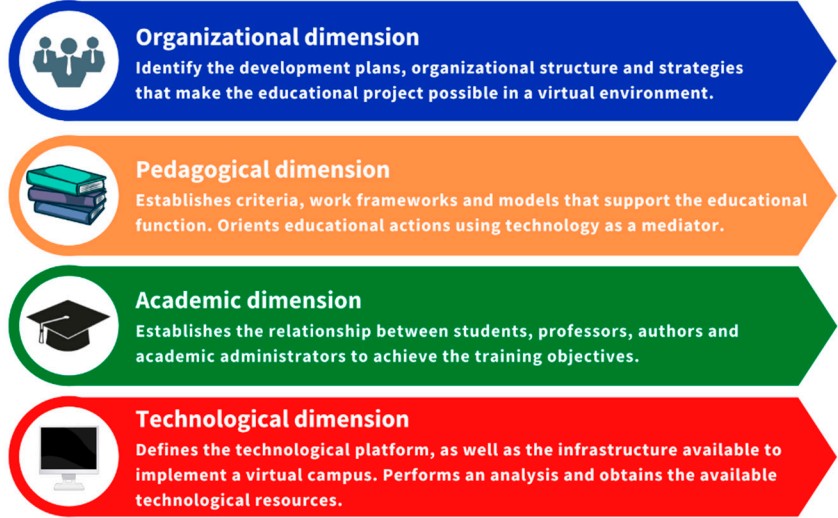

**Figure 6.** Dimensions considered through the method of implementing an accessible virtual campus.

### 5.2.2. Phase 2—Planning

This phase establishes the actions necessary to generate a strategic plan of action with the objective of developing a virtual environment that covers the requirements of students with disabilities. It is recommended that one does a follow-up on the plan from the diagnosis in Phase 1. In this phase, it is suggested to represent the organizational dimension by a director and a planning phase representative in order to assess the members at different levels of the educational environment. At the end of this stage, a document detailing a strategic plan is structured by dimension, priority, personnel, time constraints, success metrics, and costs.

### 5.2.3. Phase 3—Implementation

This phase executes the actions established by the strategic plan document. Regarding the organizational dimension, it is recommended that a project director is assigned with knowledge of governmental policy about inclusivity to lead the team. In this phase, the personnel for each dimension should be allocated depending on the established requirements to address. For the academic dimension it is recommended that a professor is selected with experience in accessible content, and to appoint a professional pedagogist versed in themes of inclusivity for the pedagogical dimension. Lastly, for the technological dimension, a director or ICT coordinator with knowledge of accessibility standards and technology should be appointed.

As a product of this phase, there should be a document with institutional guidelines for inclusive education policies, and plans for sensitivity and accessibility training for the entire academic community. The resulting document should also contain a pedagogic model that orients processes with inclusivity in mind, and a management system for accessible learning and accessible content.

### 5.2.4. Phase 4—Monitoring and Control

This is a transversal phase for the tracing of each action defined and taken in other phases since the beginning of the project. In the organizational dimension, the same team from Phase 2—Planning should be allocated. They would ensure that success metrics are met via checklists to organize the information reported. In case they come across a failed metric or an action faultily implemented, they should report it and issue recommendations to remedy the problem in order to obtain the expected results from the development

of an accessible virtual environment. This phase ends when all actions established in the strategic plan have been successfully executed.

The product of this phase is a document detailing accomplishment of all success metrics, and changes realized to counter failed actions.

### 5.2.5. Phase 5—Tracing

This phase is responsible for verifying compliance with the deliverables of each phase through the defined objectives, the strategic plan, and the activities carried out by those involved in each previous phase.

The tracing phase can assign new changes so that the goal is achieved, and that the checklists are complied with. By the tracing phase, it is possible to discover new issues or new elements to consider in the next iteration plan. According to the PDCA cycle, the Act stage allows the continual improvement by consolidating the external and internal stakeholders' feedback into the development of strategic plans [25].

Table 1 describes the PDCA process alongside indicators to consider in each dimension for each phase of the methodology of Hernández-Otálora et al. [63], for the implementation of an accessible virtual campus.

**Table 1.** List of criteria considered by each phase to verify the implementation of an accessible virtual campus.

| PDCA | Methodology from Otalora et al. |
| --- | --- |
| *Plan* | **Diagnosis phase** |
| | **Activity: Formation of the work team** |
| | Organizational dimension |
| | Is a professional with knowledge of governmental and institutional policies on educational inclusion included? |
| | Pedagogical dimension |
| | Is a pedagogical professional trained in accessibility issues or experience that favors the training of students with disabilities included? |
| | Academic dimension |
| | Is a member of the administrative staff knowledgeable about the processes involved in the HEI included? |
| | Technological dimension |
| | Is a member of the ICT management staff knowledgeable about standards and accessible technology included? |
| | **Activity: elaboration of the diagnostic report** |
| | **Organizational dimension** |
| | Are institutional or governmental policies that promote inclusion and/or accessibility identified? |
| | Are educational projects for persons with disabilities identified? |
| | Are institutional development plans identified that include inclusion and/or accessibility criteria? |
| | Are strategic plans that contemplate inclusion and/or accessibility criteria identified? |
| | Are institutional guidelines identified that consider the academic requirements of people with disabilities? |
| | Are technology accessibility guidelines identified? |
| | **Academic dimension** |
| | Are academic profiles that support people with disabilities identified? |
| | Are teacher training plans for people with disabilities identified? |
| | Is the technical help identified to support the academic activities of people with disabilities? |
| | Are admission, permanence and graduation processes with inclusion and accessibility criteria identified? |
| | **Pedagogical dimension** |
| | Is an academic unit in charge of complying with aspects related to the design of accessible curricula? |
| | Are pedagogical guidelines identified that promote curricular flexibility? |
| | Is there evidence of teacher training focused on the training of people with disabilities through ICT-supported techniques? |
| | Is there a bibliography in accessible formats identified? |
| | Is there accessibility sensitization and training plans for members of the academic community identified? |
| | **Technological dimension** |
| | Are accessible technological platforms for learning management identified? |

| | |
|---|---|
| | Are technical evaluation procedures for accessibility of technological platforms identified? |
| | Are accessibility elements identified in institutional portals, home pages, and virtual platforms? |
| | **Planning phase** |
| | **Activity: Development of the strategic plan** |
| | **Organizational dimension** |
| | Does the strategic plan contemplate the legal framework for accessibility? |
| | Does the strategic plan define who is responsible for the accessibility of the virtual campus? |
| | Does the strategic plan consider institutional regulations, standards, and policies that promote virtual campus accessibility? |
| | Is the strategic plan aligned with institutional development plans? |
| | Does the strategic plan consider institutional processes to manage the virtual campus? |
| | **Academic dimension** |
| | Does the strategic plan consider the socialization and sensitization of the project with the university community? |
| | Does the strategic plan consider training sessions on accessible contents for the university community? |
| | Does the strategic plan consider pedagogical mediations to guide the training processes for people with disabilities? |
| | Does the strategic plan include monitoring and advising the community on accessibility issues? |
| | **Pedagogical dimension** |
| | Does the strategic plan consider pedagogical orientation procedures for the educational attention to people with disabilities? |
| | Does the strategic plan consider the curricular revision of the programs to make the necessary adjustments from the universal design? |
| | Does the strategic plan consider training plans for human talent oriented to the development of accessibility competencies? |
| | Does the strategic plan include evaluation procedures based on the design for all principle? |
| | **Technological dimension** |
| | Does the strategic plan consider the verification of accessibility guidelines in institutional technological platforms? |
| | Does the strategic plan consider the existence of a technical aids bank? |
| | Does the strategic plan include the selection of an accessible LMS? |
| | Does the strategic plan include strategies to comply with the accessibility standards established by the W3C for technology platforms? |
| | Does the strategic plan include elements to verify the accessibility of digital content? |
| *Do* | **Implementation phase** |
| | **Activity: Implement the actions of the strategic plan** |
| | Is a permanent training program for community members implemented? |
| | Have the pedagogical actions detected in the planning phase been implemented? |
| | Was the accessible technology platform for learning management implemented? |
| | Does the content used in the technological platform comply with accessibility standards? |
| | Was the quality control mechanism for the accessible content used in the technological platform defined? |
| | Were permanent maintenance processes for the technological platform defined? |
| *Check* | **Monitoring and Control transversal phase** |
| | **Activity: supervise the implemented processes** |
| | Are the compliance indicators identified, and their percentages of compliance with them? |
| | Is the report of compliance indicators available? |
| | Are the areas of opportunity for process improvement identified? |
| *Act* | **Tracing** |
| | **Activity: Refers to monitoring the process and ensuring the responsibility defined for each phase of the methodology. The improvement plan, the objectives, as well as those responsible for executing the actions of the improvement plan are defined.** |
| | Are the dimensions involved in each stage of the improvement process identified? |
| | Are the persons responsible for each dimension identified? |
| | Are the objectives of the improvement plan defined for each responsible area? |
| | Is the improvement plan defined to be executed in the next iteration? |

## 6. Validation

The validation method performs a review by experts related to academic, technological, administrative, and research areas. The reviewers have a professional profile related to dimensions discovered in the practices to implement accessible virtual campuses on HEIs. The objective of this review is to establish the validity degree of the conceptual proposal of the process for the implementation of accessible virtual campuses.

### 6.1. Expert Reviews

The validation method by experts is defined as a tool that allows for a highly reliable judgment to be issued by means of opinions and points of view from the people who meet the profile of the established area. Through the opinion from experts, an evidence judgment and evaluation of a specific product are provided [65]. In the validation process of the proposal, a group of 16 people who fulfilled the profile related to some identified dimension and had a related role in some respects, such as research, technological development, education, among others. According to the authors [66,67], in order to ensure that the evaluation made by experts is reliable, at least five experts need to participate to be considered as a valid sample.

The recruitment method was by email notification to the experts in order to request their participation in the evaluation process. Through this notification, the executive document of the proposal was made available to the participants, as well as through the evaluation questionnaire. Both the executive document and the validation instrument are available at the following links:

[Link1]
https://docs.google.com/forms/d/e/1FAIpQLSd9h9NDYSJI9T0fysdC9p0CVvcz9iLNrAPZ
lIlZi5GRg0oWbw/viewform?usp=sf_link (accessed on 5 February 2022).
[Link2]
https://drive.google.com/drive/folders/1SL9NtiYlkPxZPwQVcTH8bcIjlYFzu-
MOU?usp=sharing (accessed on 5 February 2022).

In order to obtain results, a conceptual evaluation model proposed by Mora, M [68] was applied. This instrument consists of eight questions (numbered from Q1 to Q8), in addition to an open question, in order to obtain feedback from the participants. This instrument uses a Likert scale [69], numbered from 1 to 5, where 1 is a value of "totally disagree", and 5 is "totally agree".

As shown in Figure 7, the questions are listed according to the evaluation instrument, in addition to the percentages obtained from the answers with the experts.

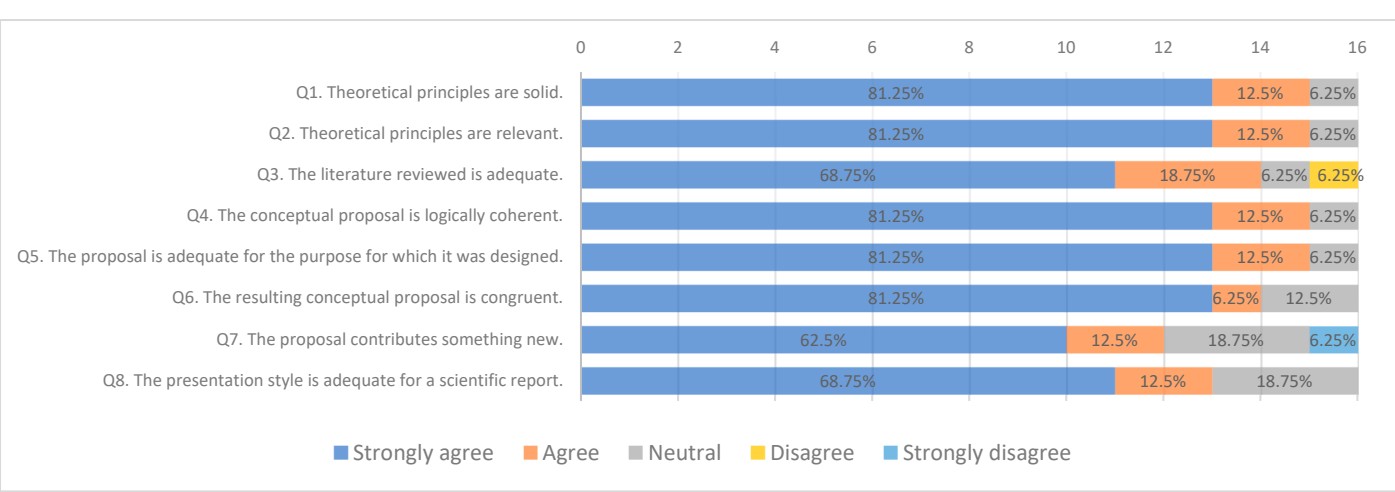

**Figure 7.** Validation results from 16 experts.

Table 2 shows the demographic information of the experts.

**Table 2.** Participants demographic information.

| ID | Gender | Country | Experience | Category |
|---|---|---|---|---|
| Expert#1 | Male | Mexico | Data research, IT research, Inclusive Technology | Industry, Evaluator |
| Expert#2 | Female | Mexico | University program of inclusive technology | Head of Department on Inclusive Technology, University administrative |
| Expert#3 | Female | Mexico | Head of career Degree in Comprehensive Development of People with Disabilities UV | Researcher, University Administrative |
| Expert#4 | Female | Mexico | Worked on the development of the academic program "Integral Development of People with Disabilities UV". Director of the School for Foreign Students UV | Academic, University Administrative |
| Expert#5 | Male | Ecuador | Accessibility researcher | Academic from Salesian Polytechnic University Ecuador |
| Expert#6 | Female | Ecuador | Accessibility researcher | Academic University from Azuay, Ecuador |
| Expert#7 | Male | Mexico | Director of Educational Innovation, UV. Worked on the development of the academic program "Integral Development of People with Disabilities UV" | University administrative |
| Expert#8 | Female | Mexico | Worked on Disability topics | Academic from Universidad Veracruzana |
| Expert#9 | Female | Mexico | Management actions for disability | Support staff Problem-Based Learning Coordination from Universidad Veracruzana |
| Expert#10 | Female | Mexico | Management actions for disability | Support staff Problem-Based Learning Coordination from Universidad Veracruzana |
| Expert#11 | Female | Mexico | Management actions for disability | Academic Coordination of Problem-Based Learning from Universidad Veracruzana |
| Expert#12 | Female | Mexico | Management actions for disability | Academic Staff Problem-Based Learning Coordination from Universidad Veracruzana |
| Expert#13 | Female | Mexico | Management actions for disability | Support staff Problem-Based Learning Coordination from Universidad Veracruzana |
| Expert#14 | Female | Mexico | Worked on Disability topics | Academic from Universidad Veracruzana |
| Expert#15 | Male | Mexico | Worked on Disability topics | Academic from Universidad Veracruzana |
| Expert#16 | Female | Mexico | Head of Academic Curriculum Development | Administrative from Universidad Veracruzana |

*6.2. Discussion of the Results with the Experts*

The collected data states that the experts agree with the proposal, where Q1 shows that 81.25% of participants have a "totally agree" perspective, while 18.75% were "agree". The theoretical principles of the proposal (Q2) are relevant to the participants, where 81.25% present a result with a value on "totally agree". The experts believe that the review of the literature carried out is adequate (Q3), 68.75% mention that they "totally agree". Most of the experts agree (81.25%) on the logical content of the proposal (Q4), which implies that it is consistent with its content. Related to the purpose of the proposal (Q5), 81.25% of the experts viewed the content as being consistent with the objective. Of the surveyed, 81.25% of the experts are totally in agreement with the proposal congruence (Q6). The novelty of the proposal (Q7) was perceived positively by the experts, with 62.5% of them totally agreeing while the rest agreed to a lesser extent.

Finally, a total of 11 experts (68.75%) considered that the proposal structure and presentation are adequate for a scientific report. This perspective was complemented by the values previously assigned by the experts.

As mentioned, part of the instrument handles an open question which allows receiving additional feedback in the form of comments or suggestions regarding the evaluation, designed proposal, among others. The comments obtained from the additional question are listed below:

1. The proposal complies with the aforementioned aspects and is understood thanks to the fact that the guidelines of the Hernández-Otálora and PDCA methodology are well established.
2. The proposal contributes to accessibility; its scientific support is adequate as well as being coherent and having a sufficient design for a scientific report.
3. It is important that the methodology encompasses the dimensions described to fortify a final result.
4. A SWOT analysis could be considered as a starting point or annex to the initial phase that allows detecting the characteristics of HEIs in this way, it would be a good complement aimed at the teaching and student community.
5. It would be good to have a minimum and maximum score in each of the criteria of the checklist, this being a helpful parameter to observe if by applying or fulfilling some points the objective of an accessible virtual campus can be achieved.
6. Projects of this nature make it possible to reinforce the characteristics of HEIs that have been made vulnerable in some cases with the arrival of the pandemic. It is good that the methodology is reinforced with the UNESCO target on Sustainable Development points.
7. It would be good to complement the proposal with a document that allows observing a summary of the disability characteristics to which it is focused.
8. The proposal provides a good point to apply to accessible virtual campuses, however, if applicable, it would be good to reinforce this work with other types of documentation focused on disability in general and not only in educational environments to observe the importance of doing it.
9. This proposal serves as a starting point to focus an implementation process on accessible virtual campuses in an orderly manner.
10. It is understood that this document is a summary of a more extensive work, however, it presents relevant points for the implementation of accessible virtual campuses and, it would be good to take elements that are categorized within the dimensions to support researchers in your understanding.

The complete list of the experts' comments (in Spanish) is available at the following link: https://drive.google.com/drive/folders/1SL9NtiYlkPxZPwQVcTH8bcIjlYFzu-MOU?usp=sharing (accessed on 5 February 2022).

## 7. Conclusions

The search for literature through a multivocal review has allowed us to know the requirements, strategies, regulations, and status in recent years of the implementation of accessible virtual campuses. It was identified that HEIs generally have regulations in compliance with the laws adopted at the federal level in each country, in order to create equal opportunities for people with disabilities. Despite the laws or institutional policies enacted, generally there are no accessible educational environments, resulting from a lack of technical knowledge, rejection of technology, limited resources available to an HEI, or a combination of the three. These factors imply the appearance of barriers that limit the performance of students with disabilities, among which are architectural, educational, discriminatory, and pedagogical barriers.

In HEIs there are virtual platforms and institutional portals that try to support the activities of the student community; however, it was detected that generally these platforms do not comply with accessibility standards, such as those established by the W3C. In addition, the integration of virtual platforms seeks to establish processes that integrate the entry, permanence, and graduation supported by the principles of universal design, in order for people with disabilities to have equal conditions.

This paper proposes a process to guide the implementation of an accessible virtual campus based on a process quality approach. This proposal is based on the PDCA continuous improvement process, which is complemented with an adapted version of the methodology for the implementation of accessible virtual environments by Hernández-Otálora et al. [63]. The process proposal includes the use of a checklist organized by phases, dimensions, activities, and criteria, which facilitates the implementation and monitoring of an accessible virtual campus by the work teams. A conceptual validation of the proposal was made by 16 experts, where 81.25% found that the proposal is logically coherent, theoretically relevant, congruent, and theoretically solid.

The usefulness of this process is conditional upon its implementation in HEIs, in order to obtain results that allow refinement of the proposal. For future research, we propose the execution of a case study that can provide feedback that contributes to obtaining results to improve the process. Additionally, an evaluation method is required in order to assess the degree of process completeness based on the analysis of the criteria and their relevance.

**Author Contributions:** Conceptualization, F.S.V.; Investigation, G.C.V.; Methodology, J.C.P.-A.; Supervision, S.L.-M. and S.O.T.; Validation, S.L.-M.; Writing—original draft, F.S.V.; Writing—review & editing, J.C.P.-A. All authors have read and agreed to the published version of the manuscript.

**Funding:** This research work has been co-funded by the Erasmus+ Program of the European Union, project EduTech (609785-EPP-1-2019-1-ES-EPPKA2-CBHE-JP).

**Institutional Review Board Statement:** Not applicable.

**Informed Consent Statement:** Informed consent was obtained from all subjects in-volved in this study.

**Data Availability Statement:** The survey used in this study is available at URL: https://docs.google.com/forms/d/e/1FAIpQLSd9h9NDYSJI9T0fysdC9p0CVvcz9iLNrAPZlIlZi5GRg0oWbw/viewform?usp=sf_link, accessed on 5 February 2022. Data related to the expert evaluation are available at URL: https://drive.google.com/drive/folders/1SL9NtiYlkPxZPwQVcTH8bcIjlYFzu-MOU?usp=sharing, accessed on 5 February 2022.

**Acknowledgments:** This deliverable has been co-funded by the Erasmus+ Program of the European Union, project EduTech (609785-EPP-1-2019-1-ES-EPPKA2-CBHE-JP). The European Commission's support of the production of this publication does not constitute an endorsement of the contents, which reflect the views only of the authors, and the Commission cannot be held responsible for any use which may be made of the information contained therein.

**Conflicts of Interest:** The authors declare no conflict of interest.

## Appendix A

**Table A1.** Studies selected in white literature. From Vázquez et al. [28].

| ID | Title of Article |
|---|---|
| S01 | McAlvage, K., & Rice, M.F. Access and Accessibility in Online Learning: Issues in Higher Education and K-12 Contexts. *From "OLC Outlook: An Environmental Scan of the Digital Learning Landscape"*. (**2018**)**.** |
| S02 | Konecki, Mario et al. "Accessible data visualization in higher education." *41st International Convention on Information and Communication Technology, Electronics and Microelectronics (MIPRO)* (**2018**): 0733–0737. |
| S03 | Romero-Chacón, Víctor et al. "Adapting SCRUM Methodology to Develop Accessible Web Sites." *International Conference on Inclusive Technologies and Education (CONTIE)* (**2019**): 112–1124. |
| S04 | Esterking, Ana Elena; González, Juana B.; Chávez, María Gabriela "Análisis de las trayectorias educativas de los alumnos con discapacidad en la Universidad Nacional de Tucumán:" [**Analysis of educational trajectories of students with disabilities at the Universidad Nacional del Tucumán**]. *En: Revista RUEDES, Año 5, no. 7, p. 19–38*. **2016**. Available online: https://bdigital.uncu.edu.ar/8395. |
| S05 | Putnam, Cynthia et al. "Best Practices for Teaching Accessibility in University Classrooms: Cultivating Awareness, Understanding, and Appreciation for Diverse Users." *ACM Trans. Access. Comput. 8* (**2016**): 13:1–13:26. |
| S06 | Gutiérrez Mozo, M. E., et al. "Campus Inclusivo, Campus Tecnológico". [**Inclusive Campus, Technological Campus**] En: Roig-Vila, Rosabel (coord.). Memorias del Programa de Redes-I3CE de calidad, innovación e investigación en docencia universitaria. Convocatoria 2017-18 = *Memòries del Programa de Xarxes-I3CE de qualitat, innovació i investigació en docència universitària. Convocatòria 2017–18. Alicante: Universidad de Alicante, Instituto de Ciencias de la Educación* (ICE), **2018**. ISBN 978-84-09-07041-1, pp. 2815–2834 |
| S07 | Schmetzke, Axel et al. "Collection Development, E-Resources, and Meeting the Needs of People with Disabilities." (**2015**). |
| S08 | Vargas, Sarmiento et al. "El Programa de Discapacidad de la Facultad de Filosofía y Letras. El desafío de comprometernos con la inclusión: avances y perspectivas." [**The Disability Program of the Faculty of Philosophy and Letters. The challenge of committing ourselves to inclusion: advances and perspectives**] (**2017**). |
| S09 | Castaño Mesa. El ser humano en situación de discapacidad incluido en la educación superior: avances en el contexto colombiano. [**The human being in a situation of disability included in the higher education: advances in the colombian context**] (**2015**). Available online: https://repository.uniminuto.edu/handle/10656/5353. |
| S10 | Behm, Gary W. et al. "Enhancing Accessibility of Engineering Lectures for Deaf & Hard of Hearing (DHH): Real-time Tracking Text Displays (RTTD) in Classrooms." (**2015**). |
| S11 | Joza. Estudio de caso de un estudiante con discapacidad visual en educación superior [**Case study of a student with visual impairment in higher education**]. (**2016**). Available online: https://repositorio.pucese.edu.ec/handle/123456789/746. |
| S12 | Rivas-Pérez, Tribeth et al. "EULER—Mathematical Editing by Voice Input for People with Visual Impairment." *2019 International Conference on Inclusive Technologies and Education (CONTIE)* (**2019**): 9–95. |

| S13 | Kearney-Volpe, Claire et al. "Evaluating Instructor Strategy and Student Learning Through Digital Accessibility Course Enhancements." The 21st International ACM SIGACCESS Conference on Computers and Accessibility (**2019**): n. pag. |
|-----|---|
| S14 | Chiou, Paul T. and Gilbert S. Young. "Implementing Recommendations of Accessibility Technology Guidelines—The Quantitative Effects and Benefits it Offers to Non-disabled Students." 2017 *International Conference on Computational Science and Computational Intelligence (CSCI)* (**2017**): 1137–1142. |
| S15 | Berrios & Mena. Inclusión de estudiantes con discapacidad en la educación Superior. [**Inclusion of students with disabilities in higher education**]. Revista Espacios. (**2012**) Available online: https://www.revistaespacios.com/a18v39n49/a18v39n49p06.pdf (accessed on 14 January 2022). |
| S16 | García, Carlos et al. "La accesibilidad como derecho: desafíos en torno a nuevas formas de habitar la Universidad." [**Accessibility as a right: challenges around new ways of inhabiting the University**] (**2015**). |
| S17 | García & Barredo. La Accesibilidad Universal en la Educación Superior Online. Caso: Universidad Isabel I. [**Universal Accessibility in Higher Education On-line. Case: Isabel I University**]. (**2019**). Available online: https://experiencias.ecci.edu.co/LibroExperienciasSignificativasVII.pdf (accessed on 17 January 2022). |
| S18 | Jara Cobos. La inclusión socioeducativa en la comunidad universitaria: perspectivas y desafíos de la educación superior en ecuador y en España. [**Socio-educational inclusion in the university community: prospects and challenges of higher education in ecuador and spain**] (**2015**). Available online: https://redib.org/Record/oai_articulo1168765-la-inclusi%C3%B3n-socioeducativa-en-la-comunidad-universitaria-perspectivas-y-desaf%C3%ADos-de-la-educaci%C3%B3n-superior-en-ecuador-y-en-espa%C3%B1a (accessed on 24 January 2022) |
| S19 | Campos Lazaro & Canelo Pacheco. Las barreras que limitan la educación inclusiva y su relación con el rendimiento académico de los estudiantes con discapacidad, de la Universidad Nacional de San Agustín de Arequipa-2018. [**The barriers that limit inclusive education and its relationship with the academic performance of students with disabilities, from the National University of San Agustín de Arequipa-2018**]. (**2019**). Available online: http://repositorio.unsa.edu.pe/handle/UNSA/9395 (accessed on 14 January 2022) |
| S20 | Repa, Melissa Jayne. "Leadership to support e-quality for all: a study of a systemwide accessible technology policy implementation." (**2015**). |
| S21 | Cinotti et al. Manual de Formación [**Trainning Manual**]. (**2015**). Available online: http://docplayer.es/184005441-Manual-de-formacion-edicion-de-alessia-cinotti-giulia-righini-y-roberta-caldin-alma-mater-studiorum-universita-di-bologna.html (accessed on 20 January 2022) |
| S22 | Benlloch, José-Vicente et al. "Marketing EIE programmes in higher education towards students from underrepresented groups." *2015 International Conference on Information Technology Based Higher Education and Training (ITHET)* (**2015**): 1–6. |
| S23 | Béjar, Rocío Molina. "Responsabilidad social de las instituciones de educación superior (IES) frente a la educación inclusiva de personas con discapacidad." [**Social responsibility of higher education institutions (HEIs) in the inclusive education of people with disabilities**] (**2016**). |
| S24 | Jauregui, R. B. and María del Pilar. "Responsabilidad social universitaria frente a las dificultades específicas del aprendizaje." [**University social responsibility in the face of specific learning difficulties**] (**2019**). |

| | |
|---|---|
| S25 | Ribeiro, Sandra Aparecida Benite et al. "University Placement Test: A proposal to decrease evasion and retention." Itinerarius Reflectionis (**2018**): n. pag. |

## Appendix B

**Table A2.** List of studies in grey literature. From Vázquez et al [28].

| ID | Title of Article |
|---|---|
| S26 | California Polytechnic State University. Access For All. (**2017**) Available online: https://ctlt.calpoly.edu/access-for-all (accessed on 2 January 2022). |
| S27 | Bellingham Technical College. Accessibility (**2017**). Available online: https://www.portofbellingham.com/940/Accessibility (accessed on 2 January 2022). |
| S28 | University of Minnesota. Accessibility of Information Technology (**2018**). Available online: https://policy.umn.edu/it/webaccess (accessed on 2 January 2022). |
| S29 | California State Polytechnic University. Accessibility Standards. (**2018**) Available online: https://www.cde.ca.gov/re/di/ws/accessibility.asp (accessed on 2 January 2022). |
| S30 | Renton Technical College. Accessible Technology (**2016**). Available online: https://rtc.edu/accessibility (accessed on 2 January 2022). |
| S31 | Seattle Colleges. Accessible Technology at Seattle Colleges. (**2017**). Available online: https://www.seattlecolleges.edu/about/accessible-technology-seattle-colleges (accessed on 2 January 2022). |
| S32 | Santa Monica College. Accessible technology at SMC. (**2016**). Available online: https://www.smc.edu/student-support/center-for-students-with-disabilities/accessible-technology/ (accessed on 2 January 2022). |
| S33 | Marquette University. Accessible technology policy. (**2017**). Available online: https://www.marquette.edu/accessible-technology/accessible-technology-policy.php (accessed on 2 January 2022). |
| S34 | Indiana University. An Overview of Compliance at IU Online. (**2018**). Available online: https://teachingonline.iu.edu/about/index.html (accessed on 2 January 2022). |
| S35 | Martínez Maldonado. Aulas Abiertas Especializadas: aspectos a tener en cuenta para promover una Educación Inclusiva. [Qualitative study on Specialized Open Classrooms: contributions to the center, teachers and students] (**2017**). Available online: https://revistaprismasocial.es/article/view/4248 (accessed on 2 January 2022). |
| S36 | Falloon. Best Practices in Accessibility for Purchasing and Marketing E-Resources: Purchasing and VPAT & GPAT Statements. (**2019**). Available online: https://guides.cuny.edu/c.php?g=393890&p=3167772 (accessed on 2 January 2022). |
| S37 | EDUCAUSE. Building a Culture of Accessibility in Higher Education. (**2018**). Available online: https://er.educause.edu/blogs/2018/7/building-a-culture-of-accessibility-in-higher-education (accessed on 20 January 2022). |
| S38 | Centralia College. Consumer Information Disclosures. (**2018**). Available online: https://www.centralia.edu/about/disclosures.aspx (accessed on 12 January 2022). |
| S39 | Gavilanes Guairacaja. El Derecho a la Educación Superior de Personas con Discapacidad en la Universidad Central del Ecuador, Carrera de Derecho dentro del periodo académico 2015 (**2018**). Available online: http://www.dspace.uce.edu.ec/handle/25000/14234 (accessed on January 14 2022). |

| | |
|---|---|
| S40 | Eastern Washington University. EWU 402-03 Accommodating Persons with Disabilities (**2017**). Available online: https://inside.ewu.edu/policies/knowledge-base/ewu-402-03-accommodating-persons-with-disabilities-2/ (accessed on 21 January 2022). |
| S41 | Universidad de Córdoba. Guía Universitaria para estudiantes con discapacidad. [**University Guide for students with disabilities**] (**2015**). Available online: https://www.fundacionuniversia.net/content/dam/fundacionuniversia/pdf/guias/Atencion-a-la-discapacidad_2015.pdf (accessed on 24 January 2022). |
| S42 | National Federation of the Blind. Higher Education Accessibility Online Resource Center (**2016**). Available online: https://nfb.org/programs-services/center-excellence-nonvisual-access/higher-education-accessibility-online-resource (accessed on 27 January 2022). |
| S43 | CNDH. Informe Especial sobre el Derecho a la Accesibilidad de las personas con discapacidad. [**Special Report on the Right to Accessibility of Persons with Disabilities**] (**2019**). Available online: https://www.cndh.org.mx/documento/informe-especial-sobre-el-derecho-la-accesibilidad-de-las-personas-con-discapacidad (accessed on 22 January 2022). |
| S44 | Observatorio de la Discapacidad. La discapacidad en la agenda de la I+D+i en España. [**Disability on the I+D+I agenda in Spain**] (**2018**). Available online: https://www.observatoriodeladiscapacidad.info/la-discapacidad-en-la-agenda-de-la-idi-en-espana/ (accessed on 6 January 2022). |
| S45 | University of Washington. Legal Cases by Issue. (**2015**). Available online: https://www.washington.edu/accessibility/requirements/legal-cases-by-issue/ (accessed on 14 January 2022). |
| S46 | LexJuris de Puerto Rico. Ley Núm. 171 de 2016. [**Law Number 171-2016**] (**2016**). Available online: https://www.lexjuris.com/lexlex/Leyes2016/lexl2016171.htm (accessed on 20 January 2022). |
| S47 | Wilson, K. Meeting the Accessibility Needs of Adult Students in Online Classes (**2016**). Available online: https://unbound.upcea.edu/innovation/contemporary-learners/meeting-the-accessibility-needs-of-adult-students-in-online-classes/ (accessed on 12 January 2022). |
| S48 | Best Colleges. Overview of college resources for students with disabilities. (**2019**). Available online: https://www.bestcolleges.com/resources/students-with-disabilities/ (accessed on 26 January 2022). |
| S49 | CEDD. Plan de Acción de la Estrategia Española sobre Discapacidad 2014–2020 (Centro Español de Documentación sobre Discapacidad. [**Action Plan of the Spanish Strategy on Disability 2014–2020**] (**2015**). Available online: https://www.observatoriodeladiscapacidad.info/wp-content/uploads/2016/12/Informe-Eval-Fase1-PAEED-OED-web.pdf (accessed on 25 January 2022). |
| S50 | San Jose State University. Policies. (**2019**). Available online: https://catalog.sjsu.edu/content.php?catoid=12&navoid=4148 (accessed on 12 January 2022). |
| S51 | Louisiana Tech University. Policy 1433—Americans with Disabilities Policy. (**2016**). Available online: https://www.latech.edu/administration/policies/p-1433/ (accessed on 27 January 2022). |
| S52 | Northern Illinois University. Policy on Purchasing, Developing, Maintaining and Using Accessible Electronic and Information Technology (EIT). (**2018**). Available online: https://www.niu.edu/ethics-compliance/technology-accessibility/accessible-eit-policy.shtml (accessed on 12 January 2022). |

| | |
|---|---|
| S53 | Universidad Popular del César. Políticas de educación inclusiva de la Universidad Popular del Cesar. [**Inclusive education policies of the Popular University of Cesar.**] (**2016**). Available online: https://www.unicesar.edu.co/index.php/es/normatividad/doc_download/3163-acuerdo-no-047-del-26-de-agosto-de-2016-anexo-politica-educacion-inclusiva (accessed on 22 January 2022). |
| S54 | Universidad de Granada. Propuesta de normativa para la atención al estudiante con discapacidad y otras necesidades específicas de apoyo educativo. [**Proposal of regulations for the attention to students with disabilities and other specific needs of educational support**] (**2016**). Available online: https://www.ugr.es/sites/default/files/2017-09/NCG1114.pdf (accessed on 2 January 2022). |

## Appendix C

**Table A3.** List of studies in white literature for Section 3.

| ID | Title of Article |
|---|---|
| S55 | Batanero-Ochaita, Concepcion & de-Marcos, Luis & Rivera, Luis & Holvikivi, Jaana & Hilera, José & Otón, Salvador & Rivera-Galicia, Luis. Improving Accessibility in Online Education: Comparative Analysis of Attitudes of Blind and Deaf Students Toward an Adapted Learning *Platform. IEEE Access*. (**2021**) PP. 1–1. |
| S56 | Sanchez-Gordon, Sandra & Aguilar-Mayanquer, Carmen & Calle-Jimenez, Tania. Model for Profiling Users With Disabilities on e-Learning *Platforms. IEEE Access*. (**2021**) PP. 1–1. |
| S57 | Nieves, Liliana & Crisol Moya, Emilio & Montes, Rosana. A MOOC on universal design for learning designed based on the UDL paradigm. *Australasian Journal of Educational Technology*. (**2019**) *35*. 30–47. |

## Appendix D

**Table A4.** List of studies in grey literature for Section 3.

| ID | Title of Article |
|---|---|
| S58 | O'Keefe, L., Rafferty, J., Gunder, A., Vignare, K. Delivering high-quality instruction online in response to COVID-19: Faculty playbook. *Every Learner Everywhere*. (**2020**). |
| S59 | Chanco, Cristhian & Moquillaza, Arturo & Diaz, Ediber & Paz, Freddy. Usability and Accessibility Evaluation of the Virtual Campus of a Peruvian University through the Use of a Mobile Phone. (**2019**). |
| S60 | Ingavélez-Guerra, Paola et al. "Automatic Adaptation of Open Educational Resources: An Approach From a Multilevel Methodology Based on Students' Preferences, Educational Special Needs, Artificial Intelligence and Accessibility Metadata." IEEE Access 10 (**2022**): 9703–9716. |
| S61 | Rice, Mary & Ortiz, Kelsey. Perceptions of Accessibility in Online Course Materials: A Survey of Teachers from Six Virtual Schools. (**2020**) 6. 245–264. |

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
