# Peer review of "Towards the Implementation Process of Accessible Virtual Campuses in Higher Education Institutions in Latin America"

_applsci, doi:10.3390/app12115470_

Round 1

Reviewer 1 Report

lines 46-50
- The argumentation is not clear. The cited Molina-Perez states limitations for implementing IT (in education). 
- Deriving the need for"a mechanism to determine the technological accessibility needs" might be justifiable. However, it is questionable if it "ensures positive results" (of what?).

The sentence lines 105-108, ending "..contributes to guarantee equitable access for all IUCEA" 
leaves asking, why IUCEA - and only IUCEA? The sentence seems to point out generally accepted principles of good education management.

The research approach seems to be consultative, taking well-established, renowned methodologies (neither PDCA nor the generic implementation process distinct 

lines 112-113: is the preposition "from" correct, and the intended meaning?  

2.3 Answers to research questions (in this heading, it would be good to remind that this is about the "2. Technological accessibility needs in higher education institutions" and "2.1 Identification of needs in environments related to HEIs"

- The review target and analysis is centred around "needs". 
Are studies included that would (empirically) address the needs of the impaired directly? 
The literature base seems rather be about institutional and (inter)national practices, policies, norms/standards and laws regarding HEIs and their activities. 

An example, line 215: 
2.3.4 NRQ3: What are the needs addressed in technological accessibility strategies implemented by HEIs?
-> The beginning of the text paragraph presents first rather technical details / "tactics", not institutional strategies; Depends, of course, your interpretation of the term 'strategy' - the following do represent institutional principles / good practices, that could be called strategies

You have twice the same subsection heading 
Line 189 AND 215 read:
"2.3.3 NRQ3: What are the needs addressed in technological accessibility strategies 189
implemented by HEIs?"

The authors would do well to reconsider section 2 
- If it is not possible to find literature presenting results of efforts eliciting the needs, then revise the target of this review. 
- Further, the section would greatly benefit from a reorganization of the content to provide conceptually clear, distinct aspects as the results of the analysis of this literature: What are the activity level "needs" of students, which requirements does this introduce the different aspects of HEIs and their stakeholder groups?

Line 237 Figure 2. Thematic map of association of terms in response to the research questions [43].

This mapping of the "themes" is interesting. However, the analysis from which it is a result is not transparent (cf. the weaknesses, as mentioned earlier found in Section 2). 

The three groups of barriers appear plausible, however not clear how they are achieved "Based on the analysis of the information contained within the selected studies" 
These barriers are quite different in nature, requiring different research approaches to find out (thinking about attitudes vs architectures). 
How the 'pedagogical barriers' are described, the issues appear to be not only about pedagogy but also about other aspects of HEI human resource management. 

Line 232 "the method of Cruzes et al. [43]" - what method is this?  

Section 3 sets the target to identify (example? best practice?) cases of accessible virtual campuses.

It is not a good practice to have identical section headings (3, 3.1 Accessible virtual campuses) -> the distinction of RO's and RQs does not seem to add value. 

Another literature review follows. The literature is searched in research literature bases/portals. 
At this point, the question could be raised, what about exemplary implementations without empirical research on them, i.e. not to be found in a literature review?

The lack of empirics (which would strengthen this study) erodes the basis for the joint quality and development method process presented. 

Line 345, subsection

"3.3.3 RQ3: What are the needs addressed in technological accessibility strategies implemented by HEIs?"
appears to come back to the concept of "need" - 
reporting a study related to the information technologies such as this, the concept of 'requirement' would be more distinct. There are different levels to observe requirements, which in the case of this study would help to discern between "needs" arising from different levels of HEI activity, different stakeholder groups etc. The focus on the impaired students and their 'needs' (if aimed at!) is lost in the study into the national/institutional aspects. 

In IT governance, the concept of the 'needs' appears as the underlying factors influencing the requirements arising from different stakeholder groups. 

It is clear that in the case of virtual campuses, there are also other stakeholders than the impaired students. This study seems not to be consistent in the point of view it is taking. 

The suggested processes PDCA and the design&development process appear (too) broad in scope, including both institution management level concerns together with detailed aspects at the level of realization. 

As such, both the quality management cycle and the generic implementation process (of which there are innumerable variations) are valid. What about their alternatives, or the selection of just these two, for the combined process model? 

By definition, taking existing models (such as methods, process models) and applying them in a case does not necessarily create new knowledge to contribute to a field of research. 
Is their combination the original contribution of this study? Or did the combination appear already in Otalora et al. (ref. 62)? 

The items presented under the PDCA cycle and implementation process steps are "suggestions". To achieve a research result status, their validation through some external evaluation would be the minimum requirement.

From an HEI reality perspective and a quality management perspective, many more suggestions, or alternatives, of items to be included in the process steps could be brought forward. What would be the justification for the given suggestions? 

Reviewer 2 Report

This article refers to a proposal for a process that HEIs can follow to implement more accessible campuses.

The article is relatively well written, although it is recommended that it be revised to correct some language inaccuracies.

The introduction (section 1) is well written and contextualizes the study.

My main concerns are related to research design and the way the design is presented. Thus, I suggest that a new section be included between the current sections 2 and 3, in which the research design is presented, that is, that shows the reader what the research objectives are, its research questions and the methodological approach followed to get answers to research questions (research steps, data collection and processing methods, etc.). This new section will make the rest of the document clearer.

The current sections 2 and 3 describe what appear to have been two “multivocal reviews” with relatively close themes. Some comments on these two sections:

- these two sections seem to have been made by different researchers, and with different logics. I suggest that these efforts be combined and that these two sections be readjusted to present a well-structured and more coherent literature review section.

- the need to carry out multivocal reviews, which include gray literature, is not well justified. Why did you use this type of review? With what objectives?

- section 2. provides little methodological content, namely what were the objectives to be achieved in this review, what were the search strings used and in what sources, the inclusion and exclusion criteria, the number of articles obtained, etc. Section 3 includes more information and is better structured

- References in section 2 are indicated in appendix A and B, with a numbering separate from the general article references. In addition, they are sorted in alphabetical order of titles, which makes querying rather confusing.

Other aspects that need attention

- RQ3 and RQ4 of section 3.2 are the same

- Lines 115-122, last paragraph of section 1: says section II, III, etc instead of 2., 3., etc: it should be corrected

The process proposal presented in section 4 seems a little rigid and formal, it would be interesting to seek and integrate other approaches to include, in this proposal, elements that are more aligned with the current dynamics of HEIs, in which institutional and technological aspects change with increasing frequency.

Finally, I would like to see the discussion of the proposal after or at the end of section 4. As the proposal was not implemented, some questions are difficult to answer, but can be discussed prospectively: why is this proposal beneficial, or at least more beneficial than other approaches? How does it compare with other models? And so on.

Reviewer 3 Report

The article describes the proposal of a process based on the Plan-Do-Check- Act (PDCA) cycle and a methodology for the implementation of accessible learning environments, oriented to the implementation of an accessible virtual campus based on the establishment of five defined phases (diagnosis, planning, implementation, control, and tracing).

However, this proposal needs to be validated to really see its contribution.

The article does not clearly show its objective, nor does it make a study of related work, i.e., other similar proposals that may exist in that sense. 

The article is divided into three sections, but the relationship between these three sections is not clear.

Section 2 is an extract from the reference [26] written in Spanish, of which two of its authors are also authors of this article.  Although the authors refer to this paper [26], they should make it clear that it is an excerpt from that paper, including the two appendices. (This is my ethical concern).

The two appendices have references in a different format from the rest of the article and do not appear in the article (this is the same defect as in the original [26]).

The research question  in 2.3.4 is the same as that in 2.3.3

This section only takes into account publications up to December 2019. They should be updated at least until December 2021. In this section, the search mechanism followed should be made explicit, in addition to commenting that only countries in Ibero-America and the United States were taken into account.

In section 3 the authors make a search similar to the one made in [26], excluding the restriction of Ibero-America and the United States and adapting it to the search terms explained in this section.

But it is curious that they also restrict the search to documents between 2015 and 2019, when this article is written in 2022. And also that in the conclusions of this section (3.4) they return to the Latin American scope.

These are the most important shortcomings that the authors should address.

My decision is to reject the article in its current version, and recommend the authors to rewrite the article and send it for review again, where this new article should:

- clearly specify the objective and the contribution it makes with reference to other similar works.

- clearly explain the structure of the article, and the function/objective of each section.

- the bibliographic studies that need to be done (or expanded) should be up to date (as close as possible to the date of submission of the article).

- all references in the article should be in the same format as of the journal.

- if material previously published in congresses is used, it should be clearly explained, indicating how much of the content is the same and how much was added.

- if the scope of the work is Latin America, this should also be made explicit, even in the title.

- references in Spanish should have an English translation, so that the reader has a slight idea of what the article is about, since the article is not in English. In this regard, it would be good to use references that, although in Spanish, have the abstract in English.

- any proposal should be accompanied by at least a short evaluation of it.

Round 2

Reviewer 1 Report

A major concern regarding the scientific soundness is that the designed suggestion for a process model is not adequately evaluated. In research with a developing stance, i.e. designing and developing a solution to a research problem, it is essential to provide also evidence that the solution would be feasible (can be used) and also evaluative information on the solution (how would it perform in use).

This can not always be done by putting the developed solution into practice, but e.g. conducting expert interviews or testing the suggested solution in a focus group setting. The informants should be familiar with the intended use context. 
I would recommend some type of evaluation like this before releasing this paper as a research report. 

Reviewer 2 Report

The authors have revised the paper according to the comments and suggestions that were made regarding the 1st draft and have responded positively to those comments and suggestions, which was duly appreciated. I have no other objections to this article.

Reviewer 3 Report

Please, see attached file.

Round 3

Reviewer 1 Report

Please have the text proof read for language and presentation. 

Reviewer 3 Report

Thanks to authors for their work improving the paper and fulfilling  my suggestions.